# Unbiased Online Recurrent Optimization

**Corentin Tallec**
Laboratoire de Recherche en Informatique
Université Paris Sud
Gif-sur-Yvette, 91190, France
`corentin.tallec@u-psud.fr`

**Yann Ollivier**
Laboratoire de Recherche en Informatique
Université Paris Sud
Gif-sur-Yvette, 91190, France
`yann@yann-ollivier.org`

## Abstract

The novel *Unbiased Online Recurrent Optimization* (UORO) algorithm allows for online learning of general recurrent computational graphs such as recurrent network models. It works in a streaming fashion and avoids backtracking through past activations and inputs. UORO is computationally as costly as *Truncated Backpropagation Through Time* (truncated BPTT), a widespread algorithm for online learning of recurrent networks Jaeger (2002). UORO is a modification of *NoBackTrack* Ollivier et al. (2015) that bypasses the need for model sparsity and makes implementation easy in current deep learning frameworks, even for complex models. Like NoBackTrack, UORO provides unbiased gradient estimates; unbiasedness is the core hypothesis in stochastic gradient descent theory, without which convergence to a local optimum is not guaranteed. On the contrary, truncated BPTT does not provide this property, leading to possible divergence. On synthetic tasks where truncated BPTT is shown to diverge, UORO converges. For instance, when a parameter has a positive short-term but negative long-term influence, truncated BPTT diverges unless the truncation span is very significantly longer than the intrinsic temporal range of the interactions, while UORO performs well thanks to the unbiasedness of its gradients.

Current recurrent network learning algorithms are ill-suited to online learning via a single pass through long sequences of temporal data. *Backpropagation Through Time* (BPTT Jaeger (2002)), the current standard for training recurrent architectures, is well suited to many short training sequences. Treating long sequences with BPTT requires either storing all past inputs in memory and waiting for a long time between each learning step, or arbitrarily splitting the input sequence into smaller sequences, and applying BPTT to each of those short sequences, at the cost of losing long term dependencies.

This paper introduces *Unbiased Online Recurrent Optimization* (UORO), an *online* and *memoryless* learning algorithm for recurrent architectures: UORO processes and learns from data samples sequentially, one sample at a time. Contrary to BPTT, UORO does not maintain a history of previous inputs and activations. Moreover, UORO is *scalable*: processing data samples with UORO comes at a similar computational and memory cost as just running the recurrent model on those data.

Like most neural network training algorithms, UORO relies on stochastic gradient optimization. The theory of stochastic gradient crucially relies on the unbiasedness of gradient estimates to provide convergence to a local optimum. To this end, in the footsteps of *NoBackTrack* (NBT) Ollivier et al. (2015), UORO provides provably *unbiased* gradient estimates, in a scalable, streaming fashion.

Unlike NBT, though, UORO can be easily implemented in a black-box fashion on top of an existing recurrent model in current machine learning software, without delving into the structure and code of the model.

The framework for recurrent optimization and UORO is introduced in Section 2. The final algorithm is reasonably simple (Alg. 1), but its derivation (Section 3) is more complex. In Section 6, UORO is shown to provide convergence on a set of synthetic experiments where truncated BPTT fails to display reliable convergence. An implementation of UORO is provided as supplementary material.

# 1 RELATED WORK

A widespread approach to online learning of recurrent neural networks is *Truncated Backpropagation Through Time* (truncated BPTT) Jaeger (2002), which mimics Backpropagation Through Time, but zeroes gradient flows after a fixed number of timesteps. This truncation makes gradient estimates biased; consequently, truncated BPTT does not provide any convergence guarantee. Learning is biased towards short-time dependencies. [1]. Storage of some past inputs and states is required.

Online, exact gradient computation methods have long been known (*Real Time Recurrent Learning* (RTRL) Williams & Zipser (1989); Pearlmutter (1995)), but their computational cost discards them for reasonably-sized networks.

*NoBackTrack* (NBT) Ollivier et al. (2015) also provides unbiased gradient estimates for recurrent neural networks. However, contrary to UORO, NBT cannot be applied in a blackbox fashion, making it extremely tedious to implement for complex architectures.

Other previous attempts to introduce generic online learning algorithms with a reasonable computational cost all result in biased gradient estimates. *Echo State Networks* (ESNs) Jaeger (2002); Jaeger et al. (2007) simply set to $0$ the gradients of recurrent parameters. Others, e.g., Maass et al. (2002); Steil (2004), introduce approaches resembling ESNs, but keep a partial estimate of the recurrent gradients. The original *Long Short Term Memory* algorithm Hochreiter & Schmidhuber (1997) (LSTM now refers to a particular architecture) cuts gradient flows going out of gating units to make gradient computation tractable. *Decoupled Neural Interfaces* Jaderberg et al. (2016) bootstrap truncated gradient estimates using synthetic gradients generated by feedforward neural networks. The algorithm in Movellan et al. (2002) provides zeroth-order estimates of recurrent gradients via diffusion networks; it could arguably be turned online by running randomized alternative trajectories. Generally these approaches lack a strong theoretical backing, except arguably ESNs.

# 2 BACKGROUND

UORO is a learning algorithm for recurrent computational graphs. Formally, the aim is to optimize $\theta$, a parameter controlling the evolution of a dynamical system

$$s_{t+1} = F_{\text{state}}(x_{t+1}, s_t, \theta) \tag{1}$$

$$o_{t+1} = F_{\text{out}}(x_{t+1}, s_t, \theta) \tag{2}$$

in order to minimize a total loss $\mathcal{L} := \sum_{0 \le t \le T} \ell_t(o_t, o_t^*)$, where $o_t^*$ is a target output at time $t$.

For instance, a standard recurrent neural network, with hidden state $s_t$ (preactivation values) and output $o_t$ at time $t$, is described with the update equations $F_{\text{state}}(x_{t+1}, s_t, \theta) := W_x x_{t+1} + W_s \tanh(s_t) + b$ and $F_{\text{out}}(x_{t+1}, s_t, \theta) := W_o \tanh(F_{\text{state}}(x_{t+1}, s_t, \theta)) + b_o$; here the parameter is $\theta = (W_x, W_s, b, W_o, b_o)$, and a typical loss might be $\ell_s(o_s, o_s^*) := (o_s - o_s^*)^2$.

Optimization by gradient descent is standard for neural networks. In the spirit of stochastic gradient descent, we can optimize the total loss $\mathcal{L} = \sum_{0 \le t \le T} \ell_t(o_t, o_t^*)$ one term at a time and update the parameter online at each time step via

$$\theta \leftarrow \theta - \eta_t \frac{\partial \ell_t}{\partial \theta}^\top \tag{3}$$

where $\eta_t$ is a scalar learning rate at time $t$. (Other gradient-based optimizers can also be used, once $\frac{\partial \ell_t}{\partial \theta}$ is known.) The focus is then to compute, or approximate, $\frac{\partial \ell_t}{\partial \theta}$.

BPTT computes $\frac{\partial \ell_t}{\partial \theta}$ by unfolding the network through time, and backpropagating through the unfolded network, each timestep corresponding to a layer. BPTT thus requires maintaining the full unfolded network, or, equivalently, the history of past inputs and activations. [2] *Truncated BPTT*

---

[1] Arguably, truncated BPTT might still learn some dependencies beyond its truncation range, by a mechanism similar to Echo State Networks Jaeger (2002). However, truncated BPTT's gradient estimate has a marked bias towards short-term rather than long-term dependencies, as shown in the first experiment of Section 6.

[2] Storage of past activations can be reduced, e.g. Gruslys et al. (2016). However, storage of all past inputs is necessary.

only unfolds the network for a fixed number of timesteps, reducing computational cost in online settings Jaeger (2002). This comes at the cost of biased gradients, and can prevent convergence of the gradient descent even for large truncations, as clearly exemplified in Fig. 2a.

## 3 UNBIASED ONLINE RECURRENT OPTIMIZATION

Unbiased Online Recurrent Optimization is built on top of a forward computation of the gradients, rather than backpropagation. Forward gradient computation for neural networks (RTRL) is described in Williams & Zipser (1989) and we review it in Section 3.1. The derivation of UORO follows in Section 3.2. Implementation details are given in Section 3.3. UORO's derivation is strongly connected to Ollivier et al. (2015) but differs in one critical aspect: the sparsity hypothesis made in the latter is relieved, resulting in reduced implementation complexity without any model restriction. The proof of UORO's convergence to a local optimum can be found in Massé (2017).

### 3.1 FORWARD COMPUTATION OF THE GRADIENT

Forward computation of the gradient for a recurrent model (RTRL) is directly obtained by applying the chain rule to both the loss function and the state equation (1), as follows.

Direct differentiation and application of the chain rule to $\ell_{t+1}$ yields

$$\frac{\partial \ell_{t+1}}{\partial \theta} = \frac{\partial \ell_{t+1}}{\partial o}(o_{t+1}, o_{t+1}^*) \cdot \left( \frac{\partial F_{\text{out}}}{\partial s}(x_{t+1}, s_t, \theta) \frac{\partial s_t}{\partial \theta} + \frac{\partial F_{\text{out}}}{\partial \theta}(x_{t+1}, s_t, \theta) \right). \qquad (4)$$

Here, the term $\partial s_t/\partial \theta$ represents the effect on the state at time $t$ of a change of parameter during the whole past trajectory. This term can be computed inductively from time $t$ to $t+1$. Intuitively, looking at the update equation (1), there are two contributions to $\partial s_{t+1}/\partial \theta$:

- The direct effect of a change of $\theta$ on the computation of $s_{t+1}$, given $s_t$.
- The past effect of $\theta$ on $s_t$ via the whole past trajectory.

With this in mind, differentiating (1) with respect to $\theta$ yields

$$\frac{\partial s_{t+1}}{\partial \theta} = \frac{\partial F_{\text{state}}}{\partial \theta}(x_{t+1}, s_t, \theta) + \frac{\partial F_{\text{state}}}{\partial s}(x_{t+1}, s_t, \theta) \frac{\partial s_t}{\partial \theta}. \qquad (5)$$

This gives a way to compute the derivative of the instantaneous loss without storing past history: at each time step, update $\partial s_t/\partial \theta$ from $\partial s_{t-1}/\partial \theta$, then use this quantity to directly compute $\partial \ell_{t+1}/\partial \theta$. This is how RTRL Williams & Zipser (1989) proceeds.

A huge disadvantage of RTRL is that $\partial s_t/\partial \theta$ is of size $\dim(\text{state}) \times \dim(\text{params})$. For instance, with a fully connected standard recurrent network with $n$ units, $\partial s_t/\partial \theta$ scales as $n^3$. This makes RTRL impractical for reasonably sized networks.

UORO modifies RTRL by only maintaining a scalable, rank-one, provably unbiased approximation of $\partial s_t/\partial \theta$, to reduce the memory and computational cost. This approximation takes the form $\tilde{s}_t \otimes \tilde{\theta}_t$, where $\tilde{s}_t$ is a column vector of the same dimension as $s_t$, $\tilde{\theta}_t$ is a *row* vector of the same dimension as $\theta^\top$, and $\otimes$ denotes the outer product. The resulting quantity is thus a matrix of the same size as $\partial s_t/\partial \theta$. The memory cost of storing $\tilde{s}_t$ and $\tilde{\theta}_t$ scales as $\dim(\text{state}) + \dim(\text{params})$. Thus UORO is as memory costly as simply running the network itself (which indeed requires to store the current state and parameters). The following section details how $\tilde{s}_t$ and $\tilde{\theta}_t$ are built to provide unbiasedness.

### 3.2 RANK-ONE TRICK: FROM RTRL TO UORO

Given an unbiased estimation of $\partial s_t/\partial \theta$, namely, a stochastic matrix $\tilde{G}_t$ such that $\mathbb{E}\, \tilde{G}_t = \partial s_t/\partial \theta$, unbiased estimates of $\partial \ell_{t+1}/\partial \theta$ and $\partial s_{t+1}/\partial \theta$ can be derived by plugging $\tilde{G}_t$ in (4) and (5). Unbiasedness is preserved thanks to linearity of the mean, because both (4) and (5) are affine in $\partial s_t/\partial \theta$.

Thus, assuming the existence of a rank-one unbiased approximation $\tilde{G}_t = \tilde{s}_t \otimes \tilde{\theta}_t$ at time $t$, we can plug it in (5) to obtain an unbiased approximation $\hat{G}_{t+1}$ at time $t+1$

$$\hat{G}_{t+1} = \frac{\partial F_{\text{state}}}{\partial \theta}(x_{t+1}, s_t, \theta) + \frac{\partial F_{\text{state}}}{\partial s}(x_{t+1}, s_t, \theta)\, \tilde{s}_t \otimes \tilde{\theta}_t. \tag{6}$$

However, in general this is no longer rank-one.

To transform $\hat{G}_{t+1}$ into $\tilde{G}_{t+1}$, a rank-one unbiased approximation, the following rank-one trick, introduced in Ollivier et al. (2015) is used:

**Proposition 1.** *Let $A$ be a real matrix that decomposes as*

$$A = \sum_{i=1}^{k} v_i \otimes w_i. \tag{7}$$

*Let $\nu$ be a vector of $k$ independent random signs, and $\rho$ a vector of $k$ positive numbers. Consider the rank-one matrix*

$$\tilde{A} := \left(\sum_{i=1}^{k} \rho_i \nu_i v_i\right) \otimes \left(\sum_{i=1}^{k} \frac{\nu_i w_i}{\rho_i}\right) \tag{8}$$

*Then $\tilde{A}$ is an unbiased rank-one approximation of $A$: $\mathbb{E}_\nu \tilde{A} = A$.*

The rank-one trick can be applied for any $\rho$. The choice of $\rho$ influences the variance of the approximation; choosing

$$\rho_i = \sqrt{\|w_i\| / \|v_i\|} \tag{9}$$

minimizes the variance of the approximation, $\mathbb{E}\left[\|A - \tilde{A}\|_2^2\right]$ Ollivier et al. (2015).

The UORO update is obtained by applying the rank-one trick twice to (6). First, $\frac{\partial F_{\text{state}}}{\partial \theta}(x_{t+1}, s_t, \theta)$ is reduced to a rank one matrix, without variance minimization. [3] Namely, let $\nu$ be a vector of independant random signs; then,

$$\frac{\partial F_{\text{state}}}{\partial \theta}(x_{t+1}, s_t, \theta) = \mathbb{E}_\nu\left[\nu \otimes \nu^\top \frac{\partial F_{\text{state}}}{\partial \theta}(x_{t+1}, s_t, \theta)\right]. \tag{10}$$

This results in a rank-two, unbiased estimate of $\partial s_{t+1}/\partial \theta$ by substituting (10) into (6)

$$\frac{\partial F_{\text{state}}}{\partial s}(x_{t+1}, s_t, \theta)\, \tilde{s}_t \otimes \tilde{\theta}_t + \nu \otimes \left(\nu^\top \frac{\partial F_{\text{state}}}{\partial \theta}(x_{t+1}, s_t, \theta)\right). \tag{11}$$

Applying Prop. 1 again to this rank-two estimate, with variance minimization, yields UORO's estimate $\tilde{G}_{t+1}$

$$\tilde{G}_{t+1} = \left(\rho_0 \frac{\partial F_{\text{state}}}{\partial s}(x_{t+1}, s_t, \theta)\, \tilde{s}_t + \rho_1 \nu\right) \otimes \left(\frac{\tilde{\theta}_t}{\rho_0} + \frac{\nu}{\rho_1}^\top \frac{\partial F_{\text{state}}}{\partial \theta}(x_{t+1}, s_t, \theta)\right) \tag{12}$$

which satisfies that $\mathbb{E}_\nu \tilde{G}_{t+1}$ is equal to (6). (By elementary algebra, some random signs that should appear in (12) cancel out.) Here

$$\rho_0 = \sqrt{\frac{\|\tilde{\theta}_t\|}{\|\frac{\partial F_{\text{state}}}{\partial s}(x_{t+1}, s_t, \theta)\, \tilde{s}_t\|}}, \quad \rho_1 = \sqrt{\frac{\|\nu^\top \frac{\partial F_{\text{state}}}{\partial \theta}(x_{t+1}, s_t, \theta)\|}{\|\nu\|}} \tag{13}$$

minimizes variance of the second reduction.

The unbiased estimation (12) is rank-one and can be rewritten as $\tilde{G}_{t+1} = \tilde{s}_{t+1} \otimes \tilde{\theta}_{t+1}$ with the update

$$\tilde{s}_{t+1} \leftarrow \rho_0 \frac{\partial F_{\text{state}}}{\partial s}(x_{t+1}, s_t, \theta)\, \tilde{s}_t + \rho_1 \nu \tag{14}$$

$$\tilde{\theta}_{t+1} \leftarrow \frac{\tilde{\theta}_t}{\rho_0} + \frac{\nu^\top}{\rho_1} \frac{\partial F_{\text{state}}}{\partial \theta}(x_{t+1}, s_t, \theta). \tag{15}$$

---

[3] Variance minimization is not used at this step, since computing $\sqrt{\frac{\|w_i\|}{\|v_i\|}}$ for every $i$ is not scalable.

Initially, $\partial s_0 / \partial \theta = 0$, thus $\tilde{s}_0 = 0$, $\tilde{\theta}_0 = 0$ yield an unbiased estimate at time 0. Using this initial estimate and the update rules (14)–(15), an estimate of $\partial s_t / \partial \theta$ is obtained at all subsequent times, allowing for online estimation of $\partial \ell_t / \partial \theta$. Thanks to the construction above, by induction all these estimates are unbiased. [4]

We are left to demonstrate that these update rules are scalably implementable.

### 3.3 IMPLEMENTATION

Implementing UORO requires maintaining the rank-one approximation and the corresponding gradient loss estimate. UORO's estimate of the loss gradient $\partial \ell_{t+1} / \partial_\theta$ at time $t + 1$ is expressed by plugging into (4) the rank-one approximation $\partial s_t / \partial \theta \approx \tilde{s}_t \otimes \tilde{\theta}_t$, which results in

$$\left( \frac{\partial \ell_{t+1}}{\partial o}(o_{t+1}, o_{t+1}^*) \frac{\partial F_{\text{out}}}{\partial s}(x_{t+1}, s_t, \theta) \cdot \tilde{s}_t \right) \tilde{\theta}_t + \frac{\partial \ell_{t+1}}{\partial o}(o_{t+1}, o_{t+1}^*) \frac{\partial F_{\text{out}}}{\partial \theta}(x_{t+1}, s_t, \theta). \quad (16)$$

Backpropagating $\partial \ell_{t+1} / \partial o_{t+1}$ once through $F_{\text{out}}$ returns $(\partial \ell_{t+1} / \partial o_{t+1} \cdot \partial F_{\text{out}} / \partial x_{t+1}, \ \partial \ell_{t+1} / \partial o_{t+1} \cdot \partial F_{\text{out}} / \partial s_t, \ \partial \ell_{t+1} / \partial o_{t+1} \cdot \partial F_{\text{out}} / \partial \theta)$, thus providing all necessary terms to compute (16).

Updating $\tilde{s}$ and $\tilde{\theta}$ requires applying (14)–(15) at each step. Backpropagating the vector of random signs $\nu$ once through $F_{\text{state}}$ returns $(\_, \_, \nu^\top \partial F_{\text{state}}(x_{t+1}, s_t, \theta) / \partial \theta)$, providing for (15).

Updating $\tilde{s}$ via (14) requires computing $(\partial F_{\text{state}} / \partial s_t) \cdot \tilde{s}_t$. This is computable numerically through

$$\frac{\partial F_{\text{state}}}{\partial s}(x_{t+1}, s_t, \theta) \cdot \tilde{s}_t = \lim_{\varepsilon \to 0} \frac{F_{\text{state}}(x_{t+1}, s_t + \varepsilon \, \tilde{s}_t, \theta) - F_{\text{state}}(x_{t+1}, s_t, \theta)}{\varepsilon} \quad (17)$$

computable through two applications of $F_{\text{state}}$. This operation is referred to as tangent forward propagation Simard et al. (1991) and can also often be computed algebraically.

This allows for complete implementation of one step of UORO (Alg. 1). The cost of UORO (including running the model itself) is three applications of $F_{\text{state}}$, one application of $F_{\text{out}}$, one backpropagation through $F_{\text{out}}$ and $F_{\text{state}}$, and a few elementwise operations on vectors and scalar products.

The resulting algorithm is detailed in Alg. 1. $F.\mathbf{forward}(v)$ denotes pointwise application of $F$ at point $v$, $F.\mathbf{backprop}(v, \delta o)$ backpropagation of row vector $\delta o$ through $F$ at point $v$, and $F.\mathbf{forwarddiff}(v, \delta v)$ tangent forward propagation of column vector $\delta v$ through $F$ at point $v$. Notably, $F.\mathbf{backprop}(v, \delta o)$ has the same dimension as $v^\top$, e.g. $F_{\text{out}}.\mathbf{backprop}((x_{t+1}, s_t, \theta), \delta o_{t+1})$ has three components, of the same dimensions as $x_{t+1}^\top$, $s_t^\top$ and $\theta^\top$.

The proposed update rule for stochastic gradient descent (3) can be directly adapted to other optimizers, e.g. *Adaptative Momentum* (Adam) Kingma & Ba (2014) or *Adaptative Gradient* Duchi et al. (2010). Vanilla stochastic gradient descent (SGD) and Adam are used hereafter. In Alg. 1, such optimizers are denoted by SGDOpt and the corresponding parameter update given current parameter $\theta$, gradient estimate $g_t$ and learning rate $\eta_t$ is denoted SGDOpt.$\mathbf{update}(g_t, \eta_t, \theta)$.

### 3.4 MEMORY-$T$ UORO AND RANK-$k$ UORO

The unbiased gradient estimates of UORO injects noise via $\nu$, thus requiring smaller learning rates. To reduce noise, UORO can be used on top of truncated BPTT so that recent gradients are computed exactly.

Formally, this just requires applying Algorithm 1 to a new transition function $F^T$ which is just $T$ consecutive steps of the original model $F$. Then the backpropagation operation in Algorithm 1 becomes a backpropagation over the last $T$ steps, as in truncated BPTT. The loss of one step of $F^T$ is the sum of the losses of the last $T$ steps of $F$, namely $\ell_{t+1}^{t+T} := \sum_{k=t+1}^{t+T} \ell_k$. Likewise, the forward tangent propagation is performed through $F^T$. This way, we obtain an unbiased gradient estimate in which the gradients from the last $T$ steps are computed exactly and incur no noise. The resulting algorithm is referred to as memory-$T$ UORO. Its scaling in $T$ is similar to $T$-truncated BPTT, both in

---

[4] In practice, since $\theta$ changes during learning, unbiasedness only holds exactly in the limit of small learning rates. This is not specific to UORO as it also affects RTRL.

---

**Algorithm 1** — One step of UORO (from time $t$ to $t + 1$)

**Inputs:**
- $x_{t+1}$, $o_{t+1}^*$, $s_t$ and $\theta$: input, target, previous recurrent state, and parameters
- $\tilde{s}_t$ column vector of size $state$, $\tilde{\theta}_t$ row vector of size $params$ such that $\mathbb{E}\, \tilde{s}_t \otimes \tilde{\theta}_t = \partial s_t / \partial \theta$
- SGDOpt and $\eta_{t+1}$: stochastic optimizer and its learning rate

**Outputs:**
- $\ell_{t+1}$, $s_{t+1}$ and $\theta$: loss, new recurrent state, and updated parameters
- $\tilde{s}_{t+1}$ and $\tilde{\theta}_{t+1}$ such that $\mathbb{E}\, \tilde{s}_{t+1} \otimes \tilde{\theta}_{t+1} = \partial s_{t+1} / \partial \theta$
- $\tilde{g}_{t+1}$ such that $\mathbb{E}\, \tilde{g}_{t+1} = \partial \ell_{t+1} / \partial \theta$

/* compute next state and loss */
$s_{t+1} \leftarrow F_{\text{state}}.\textbf{forward}(x_{t+1}, s_t, \theta), \quad o_{t+1} \leftarrow F_{\text{out}}.\textbf{forward}(x_{t+1}, s_t, \theta)$
$\ell_{t+1} \leftarrow \ell(o_{t+1}, o_{t+1}^*)$

/* compute gradient estimate */
$(\_, \delta s, \delta\theta) \leftarrow F_{\text{out}}.\textbf{backprop}\left((x_{t+1}, s_t, \theta), \dfrac{\partial \ell_{t+1}}{\partial o_{t+1}}\right)$
$\tilde{g}_{t+1} \leftarrow (\delta s \cdot \tilde{s}_t)\, \tilde{\theta}_t + \delta\theta$

/* prepare for reduction */
Draw $\nu$, column vector of random signs $\pm 1$ of size $state$
$\tilde{s}_{t+1} \leftarrow F_{\text{state}}.\textbf{forwarddiff}((x_{t+1}, s_t, \theta), (0, \tilde{s}_t, 0))$
$(\_, \_, \delta\theta_g) \leftarrow F_{\text{state}}.\textbf{backprop}((x_{t+1}, s_t, \theta), \nu^\top)$

/* compute normalizers */
$$\rho_0 \leftarrow \sqrt{\frac{\|\tilde{\theta}_t\|}{\|\tilde{s}_{t+1}\| + \varepsilon}} + \varepsilon, \quad \rho_1 \leftarrow \sqrt{\frac{\|\delta\theta_g\|}{\|\nu\| + \varepsilon}} + \varepsilon \text{ with } \varepsilon = 10^{-7}$$

/* reduce */
$$\tilde{s}_{t+1} \leftarrow \rho_0\, \tilde{s}_{t+1} + \rho_1\, \nu, \quad \tilde{\theta}_{t+1} \leftarrow \frac{\tilde{\theta}_t}{\rho_0} + \frac{\delta\theta_g}{\rho_1}$$

/* update $\theta$ */
SGDOpt.$\textbf{update}(\tilde{g}_{t+1}, \eta_{t+1}, \theta)$

---

terms of memory and computation. In the experiments below, memory-$T$ UORO reduced variance early on, but did not significantly impact later performance.

The noise in UORO can also be reduced by using higher-rank gradient estimates (rank-$r$ instead of rank-1), which amounts to maintaining $r$ distinct values of $\tilde{s}$ and $\tilde{\theta}$ in Algorithm 1 and averaging the resulting values of $\tilde{g}$. We did not exploit this possibility in the experiments below, although $r = 2$ visibly reduced variance in preliminary tests.

## 4 UORO'S VARIANCE IS STABLE AS TIME GOES BY

Gradient-based sequential learning on an unbounded data stream requires that the variance of the gradient estimate does not explode through time. UORO is specifically built to provide an unbiased estimate whose variance does not explode over time.

A precise statement regarding UORO's convergence and boundedness of the variance of gradients is provided in Massé (2017). Informally, when the largest eigenvalue of the differential transition operator $\partial F_{\text{state}} / \partial s$ is uniformly bounded by a constant $\delta < 1$ (which characterizes stable dynamical systems), the normalizing factors in (14) and (15) enforce that the influence of previous $\nu$'s decrease exponentially with time.

We hereby provide an experimental validation of the boundedness of UORO's variance in Fig. 1a. To monitor the variance of UORO's estimate over time, a 64-unit GRU recurrent network is trained on the first $10^7$ characters of the full works of Shakespeare using UORO. The network is then rerun

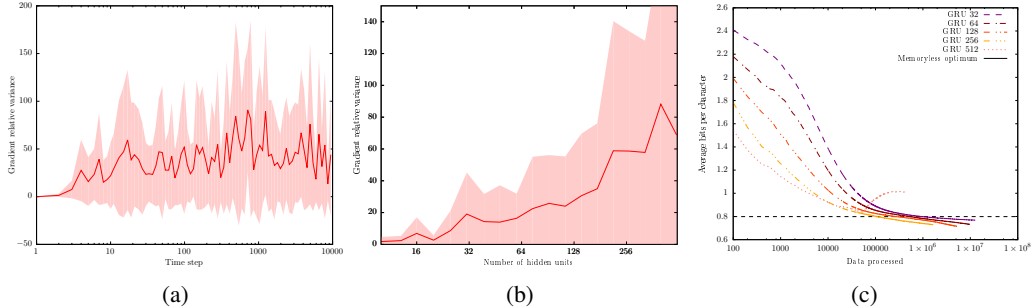

Figure 1: (a) The relative variance of UORO gradient estimates does not significantly increase with time. Note the logarithmic scale on the time axis. (b) The relative variance of UORO gradient estimates significantly increases with network size. Note the logarithmic scale on number of units. (c) Variance of larger networks affects learning on a small range copy task.

100 times on the 10000 first characters of the text, and gradients estimates at each time steps are computed, but not applied. The gradient relative variance, that is

$$\frac{\mathbb{E}\left[\|g_t - \mathbb{E}\left[g_t\right]\|^2\right]}{\|\mathbb{E}\left[g_t\right]\|^2}, \tag{18}$$

is computed, where the average is taken with respect to runs. This quantity appears to be stationary over time (Fig. 1a).

## 5 UORO'S VARIANCE INCREASES WITH THE NUMBER OF HIDDEN UNITS

As the number of hidden units in the recurrent network increases, the rank one approximation that is used to provide an unbiased gradient estimate becomes coarser. Consequently, the relative variance, as defined in (18), should increase as the number of hidden units increases.

This increase is experimentally verified in Fig. 1b. Untrained GRU networks with various number of units are run for 10 timesteps, 100 times for each size, and the UORO gradient estimate after these 10 timesteps is computed (but not applied). The relative variance of these gradients over the 100 runs is evaluated, for each network size. As shown in the figure, the relative variance increases with the number of units. Note the horizontal log scale.

The increase of the variance of the estimate with network size underlines the need for smaller learning rates when training large networks with UORO, compared to truncated backpropagation. This can imply slower learning for the kind of dependencies that truncated backpropagation can learn. The need for lower learning rates with larger networks is exemplified in Fig. 1c. GRU networks of various hidden sizes are trained with UORO on a simple copy task, as presented in Hochreiter & Schmidhuber (1997), with a lag of $T = 5$. The networks are all trained with the same decreasing learning rate, $\eta_t = \frac{10^{-4}}{1 + 3 \cdot 10^{-3} t}$. For all network sizes except the largest, the error decreases slowly but steadily. For the largest network, the variance is too large compared to the learning rate, and the error jumps sharply midway through.

## 6 EXPERIMENTS ILLUSTRATING TRUNCATION BIAS

The set of experiments below aims at displaying specific cases where the biases from truncated BPTT are likely to prevent convergence of learning. On this test set, UORO's unbiasedness provides steady convergence, highlighting the importance of unbiased estimates for general recurrent learning.

**Influence balancing.** The first test case exemplifies learning of a scalar parameter $\theta$ which has a positive influence in the short term, but a negative one in the long run. Short-sightedness of truncated algorithms results in abrupt failure, with the parameter exploding in the wrong direction, even with truncation lengths exceeding the temporal dependency range by a factor of 10 or so.

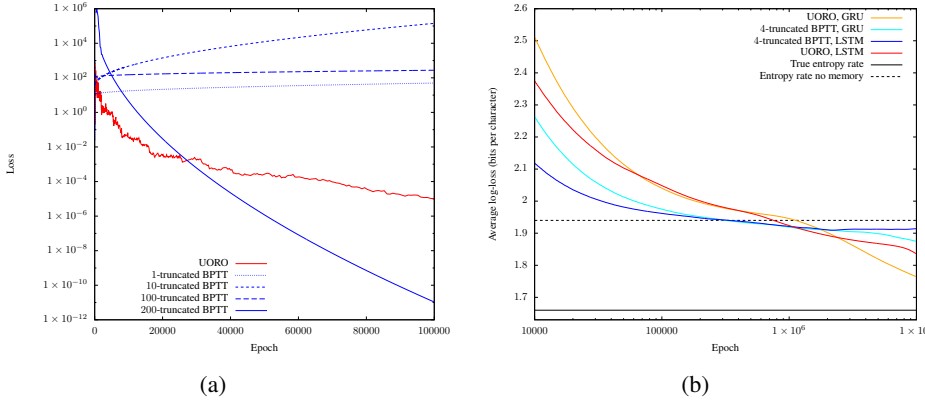

(a)                  (b)

Figure 2: (a)Results for influence balancing with 23 units and 13 minus; note the vertical log scale. (b)Learning curves on distant brackets $(1, 5, 10)$.

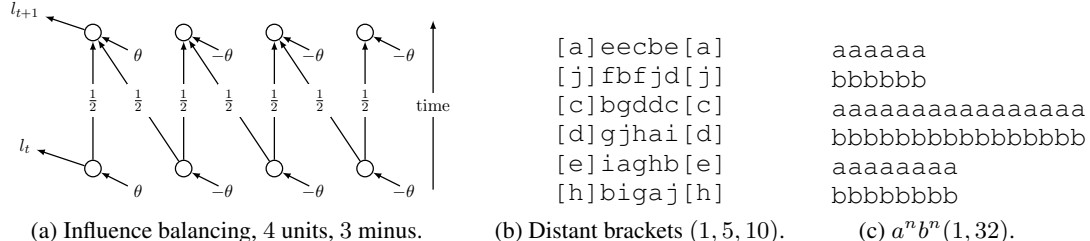

(a) Influence balancing, 4 units, 3 minus.     (b) Distant brackets $(1, 5, 10)$.     (c) $a^n b^n (1, 32)$.

Figure 3: Datasets.

Consider the linear dynamics

$$s_{t+1} = A \, s_t + (\theta, \dots, \theta, -\theta, \dots, -\theta)^\top \tag{19}$$

with $A$ a square matrix of size $n$ with $A_{i,i} = 1/2$, $A_{i,i+1} = 1/2$, and 0 elsewhere; $\theta \in \mathbb{R}$ is a scalar parameter. The second term has $p$ positive-$\theta$ entries and $n - p$ negative-$\theta$ entries. Intuitively, the effect of $\theta$ on a unit diffuses to shallower units over time (Fig. 3a). Unit $i$ only feels the effect of $\theta$ from unit $i + n$ after $n$ time steps, so the intrinsic time scale of the system is $\approx n$. The loss considered is a target on the shallowest unit $s^1$,

$$\ell_t = \tfrac{1}{2}(s_t^1 - 1)^2. \tag{20}$$

Learning is performed online with vanilla SGD, using gradient estimates either from UORO or $T$-truncated BPTT with various $T$. Learning rates are of the form $\eta_t = \frac{\eta}{1+\sqrt{t}}$ for suitable values of $\eta$.

As shown in Fig. 2a, UORO solves the problem while $T$-truncated BPTT fails to converge for any learning rate, even for truncations $T$ largely above $n$. Failure is caused by ill balancing of time dependencies: the influence of $\theta$ on the loss is estimated with the wrong sign due to truncation. For $n = 23$ units, with 13 minus signs, truncated BPTT requires a truncation $T \geq 200$ to converge.

**Next-character prediction.** The next experiment is character-level synthetic text prediction: the goal is to train a recurrent model to predict the $t + 1$-th character of a text given the first $t$ online, with a single pass on the data sequence.

A single layer of $64$ units, either GRU or LSTM, is used to output a probability vector for the next character. The cross entropy criterion is used to compute the loss. At each time $t$ we plot the cumulated loss per character on the first $t$ characters, $\frac{1}{t}\sum_{s=1}^{t} \ell_s$. (Losses for individual characters are quite noisy, as not all characters in the sequence are equally difficult to predict.) This would be the compression rate in bits per character if the models were used as online compression algorithms on the first $t$ characters. In addition, in Table 1 we report a "recent" loss on the last $100,000$ characters, which is more representative of the model at the end of learning.

Optimization was performed using Adam with the default setting $\beta_1 = 0.9$ and $\beta_2 = 0.999$, and a decreasing learning rate $\eta_t = \frac{\gamma}{1+\alpha\sqrt{t}}$, with $t$ the number of characters processed. As convergence of

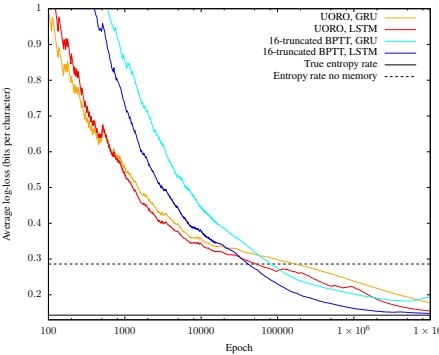 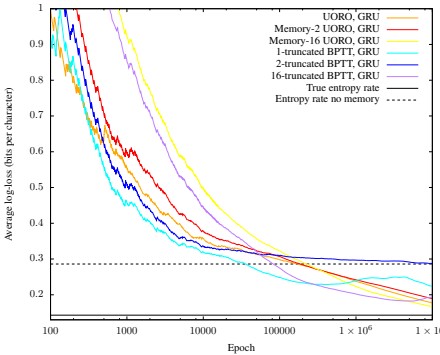

Figure 4: Learning curves on $a^n b^n_{(1,32)}$

UORO requires smaller learning rates than truncated BPTT, this favors UORO. Indeed UORO can fail to converge with non-decreasing learning rates, due to its stochastic nature.

DISTANT BRACKETS DATASET $(s, k, a)$. The distant brackets dataset is generated by repeatedly outputting a left bracket, generating $s$ random characters from an alphabet of size $a$, outputting a right bracket, generating $k$ random characters from the same alphabet, repeating the same first $s$ characters between brackets and finally outputting a line break. A sample is shown in Fig. 3b.

UORO is compared to 4-truncated BPTT. Truncation is deliberately shorter than the inherent time range of the data, to illustrate how bias can penalize learning if the inherent time range is unknown a priori. The results are given in Fig. 2b (with learning rates using $\alpha = 0.015$ and $\gamma = 10^{-3}$). UORO beats 4-truncated BPTT in the long run, and succeeds in reaching near optimal behaviour both with GRUs and LSTMs. Truncated BPTT remains stuck near a memoryless optimum with LSTMs; with GRUs it keeps learning, but at a slow rate. Still, truncated BPTT displays faster early convergence.

$a^n b^n(k, l)$ DATASET. The $a^n b^n(k, l)$ dataset tests memory and counting Gers & Schmidhuber (2001); it is generated by repeatedly picking a random number $n$ between $k$ and $l$, outputting a string of $n$ $a$'s, a line break, $n$ $b$'s, and a line break (see Fig. 3c). The difficulty lies in matching the number of $a$'s and $b$'s.

Table 1: Averaged loss on the $10^5$ last iterations on $a^n b^n(1, 32)$.

|  | Truncation | LSTM | GRU |
|---|---|---|---|
|  | No memory (default) | 0.147 | 0.155 |
| UORO | Memory-2 | 0.149 | 0.174 |
|  | Memory-16 | 0.154 | 0.149 |
|  | 1 | 0.178 | 0.231 |
| Truncated BPTT | 2 | 0.149 | 0.285 |
|  | 16 | 0.144 | 0.207 |

Plots for a few setups are given in Fig. 4. The learning rates used $\alpha = 0.03$ and $\gamma = 10^{-3}$. Numerical results at the end of training are given in Table 1. For reference, the true entropy rate is 0.14 bpc, while the entropy rate of a model that does not understand that the numbers of $a$'s and $b$'s coincide is double, 0.28 bpc.

Here, in every setup, UORO reliably converges and reaches near optimal performance. Increasing UORO's range does not significantly improve results: providing an unbiased estimate is enough to provide reliable convergence in this case. Meanwhile, truncated BPTT performs inconsistently. Notably, with GRUs, it either converges to a poor local optimum corresponding to no understanding of the temporal structure, or exhibits gradient reascent in the long run. Remarkably, with LSTMs rather than GRUs, 16-truncated BPTT reliably reaches optimal behavior on this problem even with biased gradient estimates.

## CONCLUSION

We introduced UORO, an algorithm for training recurrent neural networks in a streaming, memoryless fashion. UORO is easy to implement, and requires as little computation time as truncated

BPTT, at the cost of noise injection. Importantly, contrary to most other approaches, UORO scalably provides unbiasedness of gradient estimates. Unbiasedness is of paramount importance in the current theory of stochastic gradient descent. Furthermore, UORO is experimentally shown to benefit from its unbiasedness, converging even in cases where truncated BPTT fails to reliably achieve good results or diverges pathologically.

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
