# OpenReview forum: "Unbiased Online Recurrent Optimization"
_ICLR.cc/2018/Conference — Accept (Poster)_

### Official Review · AnonReviewer1 · 2017-11-27

**Rating:** 6
**Confidence:** 4

**Review:**

The authors introduce a novel approach to online learning of the parameters of recurrent neural networks from long sequences that overcomes the limitation of truncated backpropagation through time (BPTT) of providing biased gradient estimates.

The idea is to use a forward computation of the gradient as in Williams and Zipser (1989) with an unbiased approximation of Delta s_t/Delta theta to reduce the memory and computational cost.

The proposed approach, called UORO, is tested on a few artificial datasets.

The approach is interesting and could potentially be very useful. However, the paper lacks in providing a substantial experimental evaluation and comparison with other methods.
Rather than with truncated BPTT with smaller truncation than required, which is easy to outperform, I would have expected a comparison with some of the other methods mentioned in the Related Work Section, such as NBT, ESNs, Decoupled Neural Interfaces, etc. Also the evaluation should be extended to other challenging tasks.

I have increased the score to 6 based on the comments and revisions from the authors.

---

> ### Author Response · Authors · 2018-01-03
> **Answer to reviewer 1**
>
> Thank you for your comments and suggestions.
>
> 1/ Regarding comparison to other online methods such as NoBackTrack and Echo State Networks. For plain, fully connected RNNs, NoBackTrack and UORO turn out to be mathematically identical (though implemented quite differently), so they will perform the same. On the contrary, for LSTMs, NoBackTrack is extremely difficult to implement (to our knowledge, it has never been done); this was one of the motivations for UORO, but it makes the comparison difficult.
>
> For Echo State Networks: ESNs amount in great part to not learning the internal weights, only the output weights (together with a carefully tuned initialization). As much as we are aware, they are not known to fare particularly well on the kind of task we consider, but we may have missed relevant references.
>
> 2/ We have included a few more tasks and tests, although this remains relatively small-scale.

---

### Official Review · AnonReviewer3 · 2017-11-27
**Very interesting paper that approaches online training of RNNs in a principled way, although more experiments would make it more convincing**

**Rating:** 7
**Confidence:** 4

**Review:**

Post-rebuttal update:
I am happy with the rebuttal and therefore I will keep the score of 7.

This is a very interesting paper. Training RNN's in an online fashion (with no backpropagation through time) is one of those problems which are not well explored in the research community. And I think, this paper approaches this problem in a very principled manner. The authors proposes to use forward approach for the calculation of the gradients. The author proposes to modify RTRL by maintaining a rank one approximation of jacobian matrix (derivative of state w.r.t parameters)  which was done in NoBackTrack Paper. The way I think this paper is different from NoBackTrack Paper is that this version can be implemented in a black box fashion and hence easy to implement using current DL libraries like Pytorch.

Pros.

- Its an interesting paper, very easy to follow, and with proper literature survey.

Cons:

- The results are quite preliminary. I'll note that this is a very difficult problem.
- "The proof of UORO’s convergence to a local optimum is soon to be published Masse & Ollivier (To appear)."  I think, paper violates the anonymity.  So, I'd encourage the authors to remove this.

Some Points:

- I find the argument of stochastic gradient descent wrong (I could be wrong though). RNN's follow the markov property (wrt hidden states from previous time step and the current input) so from time step t to t+1, if you change the parameters, the hidden state at time t (and all the time steps before) would carry stale information unless until you're using something like eligibility traces from RL literature. I also don't know how to overcome this issue.

- I'd be worried about the variance in the estimate of rank one approximation. All the experiments carried out by the authors are small scale (hidden size = 64). I'm curious if authors tried experimenting with larger networks, I'd guess it wont perform well due to the high variance in the approximation. I'd like to see an experiment with hidden size  = 128/256/512/1024. My intuition is that because of high variance it would be difficult to train this network, but I could be wrong. I'm curious what the authors had to say about this.

- If the variance of the approximation is indeed high, can we use something to control the dynamics of the network which can result in less variance. Have authors thought about this ?

- I'd also like to see experiments on copying task/adding task (as these are standard experiments which are done for analysis of long term dependencies)

- I'd also like to see what effect the length of sequence has on the approximation. As small errors in approximation on each step can compound giving rise to chaotic dynamics. (small change in input => large change in output)

- I'd also like to know how using UORO changes the optimization as compared to Back-propagation through time in the sense, does the two approaches would reach same local minimum ? or is there a possibility that the former can reach "less" number of potential local minimas as compared to BPTT.


I'm tempted to give high score for this paper( Score - 7) , as it is unexplored direction in our research community, and I think this paper makes a very useful contribution to tackle this problem in a very principled way.  But I'd like some more experiments to be done (which I have mentioned above), failing to do those experiments, I'd be forced to reduce the score (to score - 5)

---

> ### Author Response · Authors · 2018-01-03
> **Answer to reviewer 3**
>
> Thank you for your insights, questions and suggestions. We have tried to attend your concerns in the revised version of the paper.
>
> 1/ As you pointed out, the results are indeed preliminary. As pointed out in the answer to Reviewer 2, it is difficult to obtain results competitive with BPTT on large scale benchmarks given the additional constraints on UORO (namely, no storage of past data, and good convergence properties, which is not the case of truncated backpropagation if dependencies exceed its truncation range).
>
> 2/ About the variance of UORO for large networks: We have added an experiment to test this. The variance of UORO does increase with netowrk size (probably sublinearly), and larger networks will require smaller learning rates.
>
> 3/ About the effect of length on the quality of the approximation: We have added an experiment to test the evolution of UORO variance when time increases along the input sequence. The variance of UORO does not explode over time, and is stationary.  A key point of UORO is the whole renormalization process (variable rho), designed precisely for this. An independent, theoretical proof for the similar case of NoBackTrack is in (Masse 2017).  Thus UORO is applicable to unbounded sequences (notably, in the experiments, datasets are fed as a single sequence, containing 10^6-10^7 characters).
>
> 4/ About the stochastic gradient descent argument: indeed one has to be careful. If UORO is used to process a number of finite training sequences, and gradient steps are performed at the end of each sequence only, then this is a fully standard SGD argument: UORO computes, in a streaming fashion, an unbiased estimate of the same gradient as BPTT for each training sequence.  However, if the gradients steps are performed at every time step, as we do here, then you are right that an additional argument is needed. The difference between applying gradients at each step and applying gradients only at the end of each sequence is at *second order* in the learning rate: if the learning rate is small, applying gradients at each time does not change the computations too much, and the SGD argument applies up to second-order terms.  This is fully formalized in (Masse 2017). If moreover, only one infinite training sequence is provided, then an additional assumption of ergodicity (decay of correlations) is needed. But in any case unbiasedness is the central property.
>
> 5/ About the optima reached by UORO vs BPTT: in the limit of very small learning rates, UORO, RTRL, and BPTT with increasing truncation lengths will all produce the same limit trajectories. The theory from (Masse 2017) proves local convergence to the *same* set of local optima for RTRL and UORO (if starting close enough to the local optimum). On the other hand, for large learning rates, we are not aware of theoretical results for any recurrent algorithm.
>
> 6/ Regarding reference (Masse 2017): this reference is now publically
> available and we provide a link in the bibliography. We were indeed aware of Masse's work a bit before it was put online, but that still covers many people, so we do not believe this breaks anonymity. Our paper is disjoint from (Masse 2017), as can be directly checked by comparing the texts.

---

### Official Review · AnonReviewer2 · 2017-11-28
**Clever trick for making general memory-efficient online unbiased RNN learning possible**

**Rating:** 8
**Confidence:** 5

**Review:**

This paper presents a generic unbiased low-rank stochastic approximation to full rank matrices that makes it possible to do online RNN training without the O(n^3) overhead of real-time recurrent learning (RTRL). This is an important and long-sought-after goal of connectionist learning and this paper presents a clear and concise description of why their method is a natural way of achieving that goal, along with experiments on classic toy RNN tasks with medium-range time dependencies for which other low-memory-overhead RNN training heuristics fail. My only major complaint with the paper is that it does not extend the method to large-scale problems on real data, for instance work from the last decade on sequence generation, speech recognition or any of the other RNN success stories that have led to their wide adoption (eg Graves 2013, Sutskever, Martens and Hinton 2011 or Graves, Mohamed and Hinton 2013). However, if the paper does achieve what it claims to achieve, I am sure that many people will soon try out UORO to see if the results are in any way comparable.

---

> ### Author Response · Authors · 2018-01-03
> **Answer to reviewer 2**
>
> Thank you for the constructive feedback. At the moment, we haven't
> succeeded in scaling UORO up to the state of the art with results competitive with backpropagation on large scale benchmarks. This may be due to the additional constraints borne by UORO, namely, both
> memorylessness and unbiasedness at all time scales.  Such datasets (notably next-character or next-word predictions) contain difficult short-term dependencies: Truncated BPTT with relatively small truncation is expected to learn those dependencies better than an algorithm like UORO, which must consider all time ranges at once.

---

### Author Response · Authors · 2018-01-03
**Paper revision**

The paper has been revised following the reviewers' advice. Two sections focusing on the evolution of the variance of the gradient approximation, both with respect to the length of the input sequence and to the size of the network have been added, along with corresponding experiments.

---

### Public Comment · ~Anirudh_Goyal1 · 2018-01-14
**Interesting Work!**

I just wanted to point out (for general readers) that this paper tries to address a very interesting research problem i.e training RNN's in an online fashion , an important open problem that has been severely under-explored in machine learning community.  Even though, the presented results are preliminary, this paper proposes a principled way to approach this problem.

Regarding the paper, I think reviewers did good job asking "right" questions, and I feel satisfied by the authors response!  I am really excited to see how can we extend this!  Good work! :-)

---

### Decision · Program_Chairs · 2018-01-29
**ICLR 2018 Conference Acceptance Decision**

**Decision:**

Accept (Poster)

**Comment:**

The reviewers agree that the proposed method is theoretically interesting, but disagree on whether it has been properly experimentally validated.   My view is that the the theoretical contribution is interesting enough to warrant inclusion in the conference, and so I will err on the side of accepting.